# The New Entity of Subacute Thyroiditis amid the COVID-19 Pandemic: From Infection to Vaccine

**DOI:** 10.3390/diagnostics12040960

**Published:** 2022-04-12

**Authors:** Mihaela Popescu, Adina Ghemigian, Corina Maria Vasile, Andrei Costache, Mara Carsote, Alice Elena Ghenea

**Affiliations:** 1Department of Endocrinology, University of Medicine and Pharmacy of Craiova, 200349 Craiova, Romania; mihaela.n.popescu99@gmail.com; 2Department of Endocrinology, Carol Davila University of Medicine and Pharmacy, 050474 Bucharest, Romania; adinaghemi@yahoo.com; 3Department of Endocrinology, C.I. Parhon National Institute of Endocrinology, 011863 Bucharest, Romania; 4Department of Paediatrics, University of Medicine and Pharmacy of Craiova, 200349 Craiova, Romania; corina.vasile93@gmail.com; 5Department of Biophysics, University of Medicine and Pharmacy of Craiova, 2003349 Craiova, Romania; costache_andrei_harper@yahoo.com; 6Department of Bacteriology-Virology-Parasitology, University of Medicine and Pharmacy of Craiova, 200349 Craiova, Romania; gaman_alice@yahoo.com

**Keywords:** new entity, endocrine, subacute thyroiditis, COVID-19 infection, coronavirus, thyroid, pandemic, COVID-19 vaccine, COVID-19 vaccination, COVID-19 immunization

## Abstract

This is a review of full-length articles strictly concerning subacute thyroiditis (SAT) in relation to the SARS-CoV-2 virus infection (SVI) and COVID-19 vaccine (COV) that were published between the 1st of March 2020 and the 21st of March 2022 in PubMed-indexed journals. A total of 161 cases were reported as follows: 81 cases of SAT–SVI (2 retrospective studies, 5 case series, and 29 case reports), 80 respective cases of SAT–COV (1 longitudinal study, 14 case series, 17 case reports; also, 1 prospective study included 12 patients, with 6 patients in each category). To our knowledge, this represents the largest cohort of reported cases until the present time. SAT–SVI was detected in adults aged between 18 and 85 years, mostly in middle-aged females. SAT–COVID-19 timing classifies SAT as viral (synchronous with infection, which is an original feature of SATs that usually follow a viral infection) and post-viral (during the recovery period or after infection, usually within 6 to 8 weeks, up to a maximum 24 weeks). The clinical spectrum has two patterns: either that accompanying a severe COVID-19 infection with multi-organ spreading (most frequent with lung involvement) or as an asymptomatic infection, with SAT being the single manifestation or the first presentation. Either way, SAT may remain unrecognized. Some data suggest that more intense neck pain, more frequent fever, and more frequent hypothyroidism at 3 months are identified when compared with non-SAT–SVI, but other authors have identified similar presentations and outcomes. Post-COVID-19 fatigue may be due to residual post-SAT hypothyroidism. The practical importance of SAT–SVI derives from the fact that thyroid hormone anomalies aggravate the general status of severe infections (particular concerns being tachycardia/arrhythmias, cardiac insufficiency, and ischemic events). If misdiagnosed, SAT results in unnecessary treatment with anti-thyroid drugs or even antibiotics for fever of unknown cause. Once recognized, SAT does not seem to require a particular approach when compared with non-COVID-19 cases, including the need for glucocorticoid therapy and the rate of permanent hypothyroidism. A complete resolution of thyroid hormone anomalies and inflammation is expected, except for cases with persistent hypothyroidism. SAT–COV follows within a few hours to a few weeks, with an average of 2 weeks (no particular pattern is related to the first or second vaccine dose). Pathogenesis includes molecular mimicry and immunoinflammatory anomalies, and some have suggested that this is part of ASIA syndrome (autoimmune/inflammatory syndrome induced by adjuvants). An alternative hypothesis to vaccine-related increased autoimmunity is vaccine-induced hyperviscosity; however, this is supported by incomplete evidence. From what we know so far concerning the risk factors, a prior episode of non-SVI–SAT is not associated with a higher risk of SAT–COV, nor is a previous history of coronavirus infection by itself. Post-vaccine SAT usually has a less severe presentation and a good outcome. Generally, the female sex is prone to developing any type of SAT. HLA susceptibility is probably related to both new types of SATs. The current low level of statistical evidence is expected to change in the future. Practitioners should be aware of SAT–COV, which does not restrict immunization protocols in any case.

## 1. Introduction

Subacute thyroiditis (SAT), also known as De Quervain or granulomatous thyroiditis, is caused by a viral infection, especially (but not exclusively) of the upper respiratory tract or by post-viral inflammation. The female/male ratio is 5/1 and the incidence is 12.1/100,000 people/year. It follows a tri-phasic clinical presentation: first—hyperthyroidism, second—transitory hypothyroidism that is remitted in 90–95% of cases, meaning the third phase—normal thyroid function—follows (except for the 5–10% of cases with permanent hypothyroidism requiring lifelong levothyroxine substitution) [1]. Typically, assessments include thyroid function anomalies, positive inflammatory markers, and highly suggestive hypoechoic ultrasound features [1]. The condition is self-limiting and non-steroidal anti-inflammatory drugs and beta-blockers, or a short course of glucocorticoid therapy for special cases, are recommended [1]. Since 2020, the entity has been listed in association with the coronavirus disease 2019 (COVID-19) infection, and since 2021 it has been related to the vaccine against the virus [1].

Our purpose is to present practical information on the COVID-19 infection and vaccination-related SAT as a new entity among cases of De Quervain thyroiditis.

## 2. Materials and Methods

This is a narrative review of the English-language literature. We included in our COVID-19-related analysis only full-length papers that were published between the 1st of March 2020 and the 21st of March 2022. Studies were identified through a PubMed-based search using the following keywords: “COVID-19 infection”, ”coronavirus”, or “COVID-19 vaccination/vaccine” in different combinations with “thyroiditis”, “thyroid”, and “thyroid disease” (see Tables 1 and 2). The inclusion criteria were clinically relevant data strictly concerning SAT in relation to either the coronavirus infection or immunization against the virus. Exclusion criteria were other types of thyroid involvement, such as destructive thyroiditis, Hashimoto’s thyroiditis, Basedow–Graves disease, etc.

The elements of statistical evidence for COVID-19-related SAT included 2 retrospective studies, 5 case series, and 29 case reports (a total of 81 patients); 80 respective cases confirmed with SAT following COVID-19 vaccination, which included 1 longitudinal study, 14 case series, 17 case reports; and also 1 prospective study that included 12 patients, with 6 patients in each category.

## 3. Results

### 3.1. Subacute Thyroiditis Induced by COVID-19 Infection

SAT might occur simultaneously with COVID-19 infection (the viral form) or it might occur within days or weeks after infection (usually within 6 to 8 weeks, although up to 6 months have been reported); the term “post-viral thyroiditis” seems more appropriate for this form [2,3]. Knowing that non-coronavirus thyroiditis follows the infection, concomitant thyroid inflammation with the triggering infection represents a particular aspect of this new entity.

The clinical picture may be associated with the typical signs and symptoms underlying hyperthyroidism and thyroid inflammation and/or various COVID-19 infection presentations (from asymptomatic forms to complicated, severe presentations including COVID-19 pneumonia and others) [4,5,6,7,8,9,10,11,12,13,14,15,16,17,18,19,20,21,22,23,24,25,26,27,28,29,30,31,32,33,34,35,36,37,38,39,40,41] (Table 1).

Some authors suggest the presence of more intense neck pain, a higher incidence of fever, and more frequent hypothyroidism at 3 months, which is opposite to other forms of SAT not associated with COVID infection [4]. However, not all the data sustain these assertions. Based on our analysis, the youngest patient was 18 years old and the oldest was 85 years old, and typical patients were middle-aged adult females [2,3,4,5,6,7,8,9,10,11,12,13,14,15,16,17,18,19,20,21,22,23,24,25,26,27,28,29,30,31,32,33,34,35,36,37,38,39,40,41]. The previous medical histories of affected individuals did not indicate any particular risk factors; a few cases with prior diagnosis of autoimmune thyroiditis or thyroid nodules/cancers were reported, but thyroid comorbidities were not associated with a higher risk of SAT in coronavirus-positive patients.

The real incidence of SAT among subjects infected with SARS-CoV-2 virus is currently unknown. The severity of COVID-19 infection that is complicated with SAT varies from severe presentations requiring hospitalization and admission to intensive care units to completely asymptomatic cases. There is one report of an 85-year-old male with postmortem diagnosis of SAT [41]. As expected, post-viral forms were less severe than viral SAT.

Based on the published data, we considered two types of SAT-associated clinical spectra. One pattern involves patients heavily affected by the COVID-19 infection and the other involves patients that develop a mildly symptomatic (or completely asymptomatic) form. Both of them may be under-diagnosed concerning SAT, for different reasons. It has been suggested that in cases with severe COVID-19 disease, SAT is difficult to recognize since multiple other manifestations of infection are present and mask thyroid inflammation, which otherwise should be listed among the extra-pulmonary spreading of the coronavirus infection [7,8]. Some authors have suggested that 20% of severe coronavirus infections have various thyroid involvements. Concerning the general COVID-19 picture of viral SAT in severe infections, the most affected organ is the lung, while the most important complication of SAT-related thyroid anomalies involves cardiac function. No relationship between hospitalization length and SAT was identified. The less severe aspects were cases of SAT as single manifestations of coronavirus infection or as first manifestation of a mild infection or associated with a clinical picture dominated by SAT rather than COVID-19 infection. Most post-viral forms also expressed this pattern. Recently, it was suggested that SAT represents an exponent of the post-COVID-19 recovery period [9,10]. Some authors have explained that post-COVID-19 fatigue induced by hypothyroidism is due to SAT after developing transitory (and probably unrecognized) hyperthyroidism [11].

Expert opinions indicate a high rate of suboptimal diagnosis of SAT since the condition is more frequent than we know. Routine COVID-19 tests are useful in each new case of SAT in the COVID-19 pandemic. Post-viral forms are more clearly identified and adequately recognized in the absence of other viral infections [2,3,4,5,6,7,8,9,10,11,12,13,14,15,16,17,18,19,20,21,22,23,24,25,26,27,28,29,30,31,32,33,34,35,36]. However, pre-pandemic versus pandemic studies analyzing SAT incidence in areas with increased incidence of COVID-19 infection did not report a higher rate but this is still an open issue [5].

Nowadays, COVID-19 infection correlates with a new entity of SAT, which is traditionally known to be associated with various viral infections (Coxsackie, influenza, mumps, adenoviruses, measles, rubella virus, etc.) [1]. The data we have from other etiologies concerning potential predisposition seem applicable to coronavirus infection; in particular, HLA (human leukocyte antigen) configuration and also female sex are more frequently affected [12,13,14]. The clear differences among other forms of SAT are yet to be established since the current amount of statistical evidence is low and thus a general conclusion is premature [15,16,17].

The pathogenesis of thyroid involvement includes virus-induced direct damage of follicular cells followed by fibrosis (mostly found in destructive thyroiditis) due to the coronavirus targeting angiotensin converting enzyme 2, which is abundant in the thyroid; immune anomalies; either increased or suppressed immune response, especially during severe infections; cytokine imbalances (for instance, TNF suppression causes increased IFN-alpha as a promotor of thyroid inflammation); thyroid inflammation may also be caused by the host immune response to coronavirus (including a post-viral reaction), not only by direct invasion [14,15,16,17].

Inflammation assays are positive in the majority of SAT cases. Based on the current data, there is no clear correlation between the clinical presentation of SAT during/after COVID-19 infection and inflammatory markers, nor was a highly suggestive marker of potential thyroid involvement identified in a patient developing coronavirus infection (except for what we already know about HLA susceptibility). IL-6 seems to be correlated with the severity of thyrotoxicosis. Thyroid hormone imbalance (low TSH, high FT4) might display the pitfall of FT3 levels (which are low in non-thyroidal illness syndrome and high in typical SAT).

The traditional evaluation based on thyroid ultrasound is also very practical and it should be routinely performed in each case of SAT considering the limits of the pandemic-related digital medicine era [18,19,20,21,22,23]. Fine-needle aspiration biopsy at the thyroid level, despite not being routinely indicated in other etiologies of De Quervain thyroiditis, in this case might become a very useful tool (and are an original feature of COVID-19-associated SAT), especially in poorly symptomatic cases, in post-viral circumstances, and in situations when differential diagnosis is difficult to establish unless a cytological report is provided [16,17].

It is important to be aware of this particular complication of the COVID-19 infection and to differentiate it from other acute complications such as pneumonia or thromboembolism as some complications require investigations where iodine contrast is used for computer tomography. Such investigations might aggravate thyroid hormone anomalies, such as, for instance, in cases with primary hyperthyroidism [6]. Also, once SAT is recognized, specific anti-thyroid drugs are unnecessary since beta-blockers and non-steroidal anti-inflammatory drugs usually control the disease; glucocorticoids should be used as a last resort for cases with severe local symptoms [24,25,26,27,28,29,30,31,32,33,34,35,36,37,38,39,40,41]. Some reported cases were already under glucocorticoid medication therapy due to COVID-19 evolution, possibly delaying the diagnosis of SAT [2,3,4,5,6,7,8,9,10,11,12,13,14,15,16,17,18,19,20,21,22,23,24,25,26,27,28,29,30,31,32,33,34,35,36,37,38,39].

We identified a single, prospective study from 2022 which included 64 patients, with 18.8% of them diagnosed with coronavirus-related SAT (3 out of 12 patients had thyroid involvement as the first sign of COVID-19 infection) and 9.3% of all (*n* = 6) diagnosed with vaccine-induced thyroiditis; the clinical presentation, laboratory assessments and therapies were similar among patients with COVID-19 versus non-COVID-19 etiologies, while SAT after immunization had statistically significantly fewer severe symptoms [39].

Overall, delaying or missing the diagnosis of SAT has at least four practical consequences. In severe forms of COVID-19 infection, thyroid disease aggravates the general status, with particular concerns being tachycardia/arrhythmias, cardiac insufficiency, and ischemic events. At the other end of the spectrum, a patient with SAT as the single manifestation of the infection may be an unrecognized case of COVID-19 and spread the disease amid the pandemic. On the other hand, there were situations involving misdiagnosis (as some of the cases from Table 1 were initially approached), where high levels of thyroid hormones were unnecessarily treated with anti-thyroid drugs or where fever of unknown cause, with or without local neck pain, was treated with antibiotics. Also, a newly detected case of hypothyroidism during the pandemic requires differential diagnosis with virus-induced low levels of thyroid hormones, including post-SAT hypothyroidism.

Once recognized, SAT does not seem to require a particular approach when compared with non-COVID-19 cases. A complete resolution of thyroid hormone anomalies and inflammation is expected; except for a small percentage of individuals with permanent hypothyroidism (the rate of which seems similar to non-COVID SAT). The current level of statistical evidence is expected to change in the next period of time.

### 3.2. Subacute Thyroiditis following Vaccine against COVID-19 Infection

SAT as a result of an immune response caused by COVID-19 vaccination represents a new topic in the medical literature due to the novelty of the vaccine [42,43,44,45,46,47,48,49,50,51,52,53,54,55,56,57,58,59,60,61,62,63,64,65,66,67,68,69,70,71,72,73,74,75,76,77,78,79]. The thyroid’s reaction is either that of SAT, with the traditional clinical presentation characterized by local anterior cervical discomfort/pain, fever, hyperthyroidism-related signs, or silent thyroiditis (the asymptomatic pattern), which is diagnosed based on blood and ultrasound assays and/or histological/cytological reports [43].

The thyroid’s involvement appears after vaccination, within a period ranging between a few hours and a few weeks (no particular pattern related to the first or second vaccine dose has been identified so far) [44]. A differential diagnosis is mandatory in order to exclude alternative causes of thyroiditis—not only concurrent COVID-19 infection itself but also influenza, cytomegalic infection, Epstein-Barr infection, measles, rubella, or even mumps [44].

The pathogenesis of vaccine-induced thyroiditis includes anomalies of immuneoinflammatory response, as similarly described in viral and post-viral forms. Immune system over-activation and molecular mimicry between the thyroid (such as thyroid peroxidase peptide sequences) and vaccine components (especially spike protein) represent the most important mechanism. Another hypothesis includes ASIA syndrome (autoimmune/auto inflammatory syndrome induced by adjuvants), suggesting that the vaccination acts as a trigger of autoimmune thyroid response, but this is yet to be determined [42,43,44,45,46,47,48,49,50,51,52,53,54,55,56,57,58,59,60,61,62,63,64,65,66,67,68,69,70,71,72,73,74,75,76,77,78,79]. An alternative theory to the vaccine-induced immune/autoimmune response is represented by vaccine-induced higher viscosity status, which may cause an abnormal increase in thyroid hormone levels due to their excessive release from the thyroid, particularly in patients displaying a higher risk for coagulation anomalies [45,46].

In vaccine-induced thyroiditis, neck pain and general changes due to inflammation may require general glucocorticoid therapy but only in selected cases and not as a general rule; hyperthyroidism–related tachycardia may be controlled with beta-blockers such as propranolol [44].

Despite these reports, vaccination against COVID-19 is encouraged since SAT is extremely rare and is typically associated with good clinical evolution and good long-term outcomes due to the condition being self-limiting [47,48].

From what we know so far concerning the risk factors, a prior episode of SAT with non-COVID 19 etiology is not associated with a higher risk of vaccine-related thyroiditis, nor is a previous history of coronavirus infection itself. Similarly, an SAT episode after the first dose of the vaccine does not restrict immunization protocols.

In general, the female sex is more prone to developing any type of SAT or autoimmune condition. Among the few reported cases, a series showed two sisters developing SAT symptoms after immunization. A specific genetic background is difficult to consider at this point (except for HLA susceptibility in non-coronavirus SAT) [49,50]. Moreover, one reported case had a previous diagnosis of Hashimoto’s thyroiditis, which traditionally does not increase the risk of subacute thyroid inflammation [49,50]. Another female subject also had a prior diagnosis of anti-thyroid antibody–induced chronic thyroiditis as well as a confirmation of papillary thyroid carcinoma which had not been treated with radioiodine therapy at the time the vaccination-induced thyroiditis was detected [51].

We also mention a reported case of (what most likely was) acute thyroiditis in association with bilateral optic neuritis after vaccination (noting that isolated optic neuritis after COVID-19 infection has also been described) [52,53].

We conclude that vaccine-related SAT is typically less symptomatic than other types and variably follows either the first or second dose within an average of 2 weeks. The diagnosis should be established in the absence of other well-known etiologies. The pathogenic mechanisms are less understood; we do not know any individual, new risk factors. Complete remission is found in all cases without specific requirements other than what we already know for SAT.

Practitioners should be aware of post-vaccine SAT, which does not represent at all a restriction for vaccination as it appears as an exceptional event based on our current level of evidence [54,55,56,57,58,59,60,61,62,63,64,65,66,67,68,69,70,71,72,73,74,75,76,77,78,79]. We clearly state that the benefits of vaccines against coronavirus far outweigh any transient issues concerning the thyroid gland (Table 2).

## 4. Discussion

### 4.1. The General Endocrine Picture of COVID-19 Infection

SAT is part of the large, heterogeneous endocrine picture described in relation to the COVID-19 infection. From an endocrinological point of view, a high risk of severe COVID-19 infection is related to the presence of diabetes mellitus, high blood pressure, obesity, Cushing’s syndrome, sleep apnea (associated with acromegaly or obesity), coagulation anomalies, glucocorticoid therapy for various conditions, etc. [80,81]. Hypocortisolemia may develop on immune grounds (in addition or not to primary/secondary hypothyroidism) due to direct or immune-mediated pituitary and adrenal lesions [82]. Recently, the potential involvement of adrenal dysfunction (following or not following prior infection-related glucocorticoid exposure) was connected with long COVID-19 syndrome [83,84]. Also, transitory impairment of spermatogenesis in males, relative and direct hypoparathyroidism, and the worsening of metabolic bone disease have also been reported in subjects experiencing SARS-CoV-2 infection [85,86,87].

### 4.2. Thyroid Workup among COVID-19 Positive Patients: Where Do We Place Subacute Thyroiditis?

Suspecting or confirming SAT in patients infected with coronavirus represents a small piece of an otherwise very complex puzzle of the thyroid’s involvement under these specific circumstances [88]. Generally, the patient with prior thyroid disease is not considered to be at a high risk of contracting COVID-19 infection, nor of developing SAT secondary to coronavirus infection or after vaccination against the virus. Recently, it was found that uncontrolled hypothyroidism rather than hyperthyroidism is more prone to developing a severe form of COVID-19 infection, especially when associated with older age (although not all authors agree), thus thyroid–COVID-19 interplay might be more complex [89,90].

It is already known that any kind of infection, including coronavirus infection, may trigger hyperthyroidism that can manifest in various ways, including severe forms like thyroid storm, and needs to be differentiated from SAT-related transitory hyperthyroidism [91,92]. The direct thyroid injury caused by the virus leads to destructive thyroiditis that is usually associated with severe circumstances such as increased cytokine pathway activation [93]. We did not include in this overview any cases with this particular type of thyroiditis. Low-T3 syndrome has been found in severely ill patients and represents a predictor for poor outcomes in individuals infected with the SARS-CoV-2 virus; SAT should be differentiated from euthyroid sick syndrome (a differential diagnosis which is easier to perform in the following weeks after coronavirus infection rather than during infection) [94,95]. On the other hand, some cases of COVID-19-related thyroiditis have been retrospectively diagnosed based on current low thyroid hormone status in patients without a prior thyroid history. It is therefore important to consider it not only as a new SAT entity but also as a new approach to the differential diagnosis of hypothyroidism amid the COVID-19 pandemic.

Some authors have suggested that the COVID-19 infection might trigger the autoimmune mechanisms underlying Hashimoto’s thyroiditis (which is traditionally connected to a large area of other endocrine and non-endocrine autoimmune diseases) [96,97,98]. Also, aggravating selenium deficiency through the infection may act as a precipitating element [99]. Currently, there is not enough statistical evidence to support this specific hypothesis despite the theoretical rationale [100,101]. Other autoimmune complications in COVID-19-positive patients also include thrombocytopenia, hemolytic anemia, Guillan-Barre syndrome, etc. [102,103,104].

The rate of hypothyroidism in coronavirus-positive patients varies from study to study by up to one-fifth, and the underlying mechanisms are mostly represented by low-T3 syndrome as well as hypophysitis-related hypothyroidism, destructive thyroiditis, primary autoimmune hypothyroidism, residual thyroid insufficiency after an episode of SAT, etc. [105]. The thyroid hormone profile in severe cases might be regarded as a surrogate prognosis tool in patients without prior thyroid conditions, according to some authors despite not being generally recognized [106,107,108]. We should also take note of patients on anti-thyroid drugs or treated with high doses of radioiodine therapy for differentiated thyroid cancer, who may find themselves in a situation with an increased risk for neutropenia and agranulocytosis, thus aggravating any type of infection; also, SAT-associated temporary excess of thyroid hormones should be differentiated from primary hyperthyroidism in order to avoid unnecessary exposure to anti-thyroid drugs [109,110]. More aggressive forms of thyroid cancer during the COVID-19 pandemic era have been suggested to be caused by the delays in evaluation and surgical treatment due to lockdown restrictions and pandemic regulations that have bottlenecked access to medical services [111,112]. The rate of thyroidectomy access for benign goiter was found by some authors to be reduced during periods where restrictions were in place [113,114]. Similarly, the extensive use of telemedicine amid the pandemic might delay the presentation of SAT. Patients with previous thyroid nodules do not appear to be at greater risk during COVID-19 infection, nor at greater risk of developing a subacute complication of the thyroid, according to what has been described so far in the literature, [115,116].

### 4.3. Endocrine Conditions and Vaccination against COVID-19 Infection

A highly debated topic in the scientific community relates to the potential events involving COVID-19 immunization, including endocrine elements. It has been suggested that vaccination might aggravate eye disease or primary hyperthyroidism in patients with a history of Basedow–Graves disease. However, the level of statistical evidence remains poor up to this moment, and currently, the most probable pathogenic link is post-vaccine increased immunogenicity (including autoimmunity exacerbation and molecular mimicry), probably in susceptible individuals, although this has yet to be determined [117,118,119]. Primary hyper functioning of the thyroid needs a clear differentiation from the transitory hyperthyroidism underlying a subacute inflammation of the gland in order to avoid anti-thyroid medication. Hypophysitis with the clinical and hormonal expression of hypopituitarism was also reported (the level of statistical evidence, however, is of case reports) [120]. Immune- and/or inflammatory-mediated endocrine elements that have developed after COVID-19 vaccination may be regarded as a part of ASIA syndrome (it has been discussed whether Graves disease is a part of the syndrome), and the risk of exacerbating the autoimmune response by the infection itself has previously been described [121,122].

Generally, we consider that vaccination against COVID-19 is not associated with particular risks in patients with prior diagnoses of different endocrine conditions. Some diseases like diabetes mellitus and adrenal insufficiency have a particular indication for vaccine prioritization due to the higher risk of infections and more severe SARS-CoV-2 evolution [123,124]. Whether SAT might follow vaccination is still an open issue; the level of current statistical evidence is low but, as many other conditions amid the pandemic, the importance of awareness comes first at this point.

Whether symptoms of SAT after a certain vaccine dose should delay the following vaccine administration is debatable and, for the moment, we not do have any particular concerns regarding protocols for immunization (but, of course, the vaccination started in 2021, so this is a limited period of time to be able to draw clear conclusions). There is only one longitudinal study, published in 2022, which included 15 cases with vaccine-associated thyroiditis of the subacute type, occurring a median of 11.5 days after immunization and a median remission time of 11.5 weeks; seven out of nine individuals were re-vaccinated and did not experience a relapse of subacute thyroiditis, while two out of nine individuals suffered an aggravation of SAT due to the second dose of the vaccine inoculation [72]. According to what we currently know, revaccination should not be restricted in patients who experienced SAT following any dose during immunization.

Overall, we recognize that the number of reported cases considering this new entity is low when compared with the millions of people that have had the coronavirus infection and received the vaccine, but many other unexpected novel clinical entities have been reported amid the pandemic at first with a low level of statistical evidence since the COVID-19 pandemic is an extraordinary, never-before-seen medical experience. This is why the lack of adequate recognition and diagnosis represents one more reason to spread the information and increase awareness in order to reduce disease-related burden if adequately recognized (including unnecessary investigations or medications) and to apply optimum health care for improved quality of life in affected patients. To our knowledge, this is the largest number of patients with either COVID-19-related or vaccine-associated SAT to date (see Table 1 and Table 2).

## 5. Conclusions

Based on the studies we have analyzed so far, 161 cases of COVID-19 infection and vaccine-related SAT have been reported. SAT in adults infected with coronavirus is a rare event that can be detected during or after infection. Early recognition improves clinical evolution and outcomes, despite the fact that the condition is self-limiting. Even though it is uncommon and still under the umbrella of low statistical evidence, practitioners should be aware of post-vaccine SAT, which does not restrict protocols for further vaccination.

## Figures and Tables

**Table 1 diagnostics-12-00960-t001:** All the cases that were published based on our previously mentioned research methodology.

Number	First Author/Reference Number	Year	Type of Study	Number of Patients	Age (Years)/Sex	Others Observations
1.	Sato D. [2]	2021	Case report	1	31/F	Asymptomatic COVID-19 infection followed by SAT after 2 w
2.	Feghali K. [3]	2021	Case report	1	41/F	Mild COVID-19 infection followed by SAT after 6 w
3.	Brancatella A. [4]	2021	Retrospective, transversal, observational study	18	av. 34 (±14)/F	Patients were included if positive COVID-19 infection within 45 d previous to SAT confirmation
4.	Pirola I. [5]	2021	Retrospective, single-center study *	1	44/M	COVID-19 pulmonary infection 7 w before SAT
5.	Ramsay N. [6]	2021	Case report	1	51/F	Mild COVID-19 infection followed by SAT after 8 d
6.	Tjønnfjord E. [7]	2021	Case report	1	“40s”/M	Mild COVID-19 infection followed by SAT after 3 w
7.	Osorio Martínez A. [9]	2021	Case report	1	64/M	COVID-19 pneumonia followed by SAT after 4 w **
8.	Whiting A. [11]	2021	Case report	1	49/M	SAT diagnosed 6 months after recovery from COVID-19 infection
9.	Reggio C. [12]	2021	Case report	1	76/M	SAT diagnosed 18 d after starting hospitalization for severe COVID-19 pneumonia ***
10.	de la Higuera López-Frías M. [13]	2021	Case report	1	36/F	SAT synchronous with COVID-19 diagnosis
11.	Ashraf S. [14]	2021	Case report	1	58/M	SAT synchronous with COVID-19 pneumonia
12.	Seyed Resuli A. [15]	2021	Case series	5	av. 30.4/F	Presentation for neck pain and odynophagia (synchronous diagnosis)
13.	Abreu R. [16]	2021	Case series	3	1.34/F	1. SATdiagnosed 28 d after COVID-19 confirmation ****
					2. 34/F	2. asymptomatic COVID-19 and SAT (routine ultrasound for diagnosis)
					3. 39/F	3. mild form of COVID-19 infection followed by SAT symptoms after 26 d *****
14.	Mathews SE. [18]	2021	Case report	1	67/M	Concurrent diagnosis of SAT and COVID-19; with acute chronic systolic heart failure on presentation
15.	Ghosh R. [19]	2021	Case report	1	50/M	SAT diagnosis after discharge for COVID-19 infection ******
16.	Davoodi L. [20]	2021	Case report	1	33/M	Concurrent diagnosis of SAT and COVID-19
17.	Khatri A. [21]	2021	Case report	1	41/F	SAT symptoms 2 w after mild COVID-19 infection resolution
18.	Sohrabpour S. [22]	2021	Case series	6	1.26/F	SAT symptoms with positive COVID-19 serology
					2.34/F	
					3.37/F	
					4.41/F	
					5.35/F	
					6.52/M	
19.	Dworakowska D. [23]	2021	Case report	1	57/F	SAT confirmation approximately 2 months after COVID-19 infection
20.	Mehmood MA. [24]	2020	Case report	1	29/F	SAT diagnosis 7 w after a mild form of COVID-19 infection
21.	Chakraborty U. [25]	2020	Case report	1	58/M	Admitted for SAT; he was positive by RT-PCR for SARS-CoV-2
22.	Álvarez Martín MC. [26]	2020	Case report	1	46/F	Admitted for SAT; she was found positive by RT-PCR for SARS-CoV-2
23.	San Juan MDJ. [27]	2020	Case report	1	47/F	Admitted for SAT; she was found positive by RT-PCR for SARS-CoV-2
24.	Ruano R. [28]	2020	Case report	1	28/F	SAT signs 4 w after first symptoms of COVID-19 infection (diarrhea)
25.	Chong WH. [29]	2020	Case report	1	37/F	Mild form of COVID-19 infection (symptoms remitted within 1 week of home isolation) followed by SAT in 4 w
26.	Campos-Barrera E. [30]	2020	Case report	1	37/F	Mild form of COVID-19 infection followed by SAT in 4 w
27.	Mattar SAM. [31]	2020	Case report	1	34/M	SAT diagnosis during hospitalization for COVID-19 (acute neck pain on 9th day of illness)
28.	Brancatella A. [32]	2020	Case series	5	Aged between 29 and 46/F	SAT developed 16–36 d after COVID-19 resolution
29.	Ruggeri RM. [33]	2021	Case report	1	43/F	SAT 6 w after the first diagnosis of COVID-19 infection
30.	Asfuroglu Kalkan E. [34]	2020	Case report	1	1/F	Admitted for SAT; she was found positive by RT-PCR for SARS-CoV-2
31.	Ippolito S. [35]	2020	Case report	1	69/F	SAT diagnosis during hospitalization for COVID-19 (day 5) *******
32.	Brancatella A. [36]	2020	Case report	1	18/F	SAT 15 d after COVID-19 confirmation (mild form)
33.	Semikov VI. [37]	2021	Case series	2	1.45/F	1. COVID-19 confirmation on day 6 after SAT onset
					2.40/F	2. SAT after 4 w COVID-19 infection with lung involvemnt
34.	Kliem T. [38]	2022	Case report	1	M	SAT plus positive serology for SARS-CoV-2 (IgG)
35.	Bahçecioğlu A.B. [39]	2022	Prospective study (*n* = 64)	12	*n* = 5 F *n* = 7 M median age = 49 y	3 out of 12 cases had SAT as first manifestation
36.	Das B. [40]	2021	Case report	1	33/M	SAT was diagnosed 1 w after PCR positve test
37.	Jakovac H. [41]	2022	Case report	1	85/M	Postmortem diagnosis

Abbreviations: SAT = subacute thyroiditis; w = week; d = days; F = female; M = male; RT-PCR = reverse transcriptase real-time qualitative polymerase chain reaction. * Between April 2020 and October 2020, *n* = 396 outpatients in an emergency; 2.5% of them diagnosed with subacute thyroiditis; among them one single case of COVID-19 related thyroiditis. ** Also associated reactive hepatitis and myocarditis at the time of subacute thyroiditis (at that time the COVID-19 pulmonary infection was remitted). *** Also associated necrotizing myopathy. **** COVID-19 assays as part of the pre-operative protocol for breast cancer. ***** All three cases had confirmation of SAT by fine-needle aspiration. ****** The patient was initially treated with carbimazole due to misdiagnosis of thyrotoxicosis and he further developed anti-thyroid arthritis syndrome. ******* The patient had a lon- time history of non-toxic benign nodular goiter.

**Table 2 diagnostics-12-00960-t002:** Evidence of subacute thyroiditis following vaccination against COVID-19 infection (the name of the vaccine as used by authors of the original studies is provided).

Number	First Author/Reference Number	Year	Type of Study	Number of Patients	Type of Vaccine	Others Observations
1.	Siolos A. [43]	2021	Case series	2	1. Pfizer-BioNTech	1. SAT (after 2 w)
					2. AstraZeneca	2. SAT (after 3 w)
2.	Kyriacou A. [44]	2021	Case report	1	Pfizer-BioNTech	SAT (after 12 h)
3.	Soltanpoor P [47]	2021	Case report	1	COVAXIN (The Bharat BiotechCOVID-19 Vaccine)	SAT (after 5 d)
4.	Saygılı ES. [48]	2021	Case report	1	CoronaVac	SAT (after 2 w)
5.	Chatzi S. [49]	2021	Case series	2	1. Pfizer-BioNTech	1. SAT (after 12 d) *
					2. Pfizer-BioNTech	2. SAT (after 4 d) **
6.	Sigstad E. [51]	2021	Case report	1	Pfizer-BioNTech	SAT (after 6 d) ***
7.	Patel KR. [54]	2021	Case report	1	NA	SAT (after 7 d)
8.	Bornemann C. [55]	2021	Case series	2	1. Spikevax (Moderna Biotech)	1. SAT (after 7 d) ****
					2. Vaxzevria (AstraZeneca)	2. ST (after 3 d) ****
9.	Oyibo SO. [56]	2021	Case report	1	ChAdOx1 nCoV-19 vaccine, (AstraZeneca)	SAT (after 7 d)
10.	Şahin-Tekin SM. [57]	2021	Case report	1	CoronaVac	SAT (after 17 d)
11.	İremli BG. [58]	2021	Case series	3	CoronaVac	SAT (after 4 to 7 d) *****
12.	Schimmel J. [59]	2021	Case series	3	BNT162B2	SAT (after 24 h)
13.	Ratnayake GM. [60]	2021	Case report	1	ChAdOx1, *Vaxzevria* (AstraZeneca)	SAT (after 2 w)
14.	Das L. [61]	2021	Case report	1	ChAdOx1 nCoV-19 (Astra Zeneca)	SAT (after 2 w)
15.	Jeeyavudeen MS. [62]	2021	Case report	1	COVID-19 mRNA vaccine BNT162b2 (Pfizer-BioNTech)	SAT (2 w after second dose)
16.	Plaza-Enriquez L. [63]	2021	Case report	1	Moderna mRNA COVID-19 vaccine	SAT (6 d after second dose)
17.	Pujol A. [64]	2021	Case series	1 out of 3 cases	Moderna mRNA COVID-19 vaccine	SAT (8 d after first dose)
18.	Sözen M. [65]	2021	Case series	4	COVID-19 mRNA vaccine (Pfizer/BioNTech^®^)	SAT onset:
						1. 1 d after first dose
						2. 6 d after second dose
						3. 4 d after first dose
						4. 20 d after second dose
						5. progressive onset after first dose and exacerbation after second dose
19.	González López J. [66]	2021	Case series	2	1. Comirnaty©	SAT was identified: 1. 3 d after second dose 2. 2 w after first dose
					2. Vaxzevria©	
20.	Khan F. [67]	2021	Case report	1	Pfizer-BioNTech	SAT (4 d after second dose)
21.	Pandya M. [68]	2021	Case series	3	1,2. Pfizer Bio-NTech	SAT onset:
						1. 10 d after first dose
					3. Moderna COVID-19 mRNA vaccine	2. 20 d after second dose
						3. 15 d after first dose
22.	Vasileiou V. [69]	2021	Case report	1	SARS-CoV-2 mRNA vaccine Comirnaty (Pfizer/BioNTech)	SAT (10 d after first dose)
23.	Pla Peris B. [70]	2021	Case series	3 out of 8 cases	Moderna^®^	SAT (10 to 14 d)
24.	Bahçecioğlu AB. [39] ******	2022	Prospective study	6 out of 64 cases	n = 4 with 2 doses of Sinovac-CoronaVac^®^ n = 2 with single dose of Pfizer-BioNtech^®^	SAT (1 to 12 w after vaccine)
25.	Bostan H. [71]	2022	Case series	2	1. Pfizer-BioNTech^®^	1. SAT after 3 d
					2. CoronaVac^®^	2. SAT after 6 d
26.	Oğuz SH. [72]	2022	Longitudinal study	15	Pfizer-BioNTech COVID-19 vaccine (BNT162b2)	SAT (after a median of 11.5 d (median time of remission 11.4 w))
27.	Jhon M. [73]	2022	Case report	1	RNA-1273 (Moderna) vaccination	SAT (5 d after first dose)
28.	Yorulmaz G. [74]	2022	Case series	11	6/11 with BNT162b2 Pfizer/BioNTech COVID-19 mRNA vaccine^®^ 4/11 with Coronavac inactivated SARS-CoV-2 vaccine^®^ 1/11 with first dose of BNT162b2 after two doses of Coronavac	SAT (after an average time of 22 d (15 to 73 d))
29.	Pipitone G. [75]	2022	Case report	1	Comirnaty by Pfizer Inc. (New York, USA)	SAT (1 w after first dose)
30.	Stasiak M. [76]	2022	Case series	2	Pfizer-BioNTech	1. SAT a few days after second dose
						2. SAT 3 w after second dose
31.	Bennet WM. [77]	2022	Case report	1	COVID-19 vaccine AstraZeneca	SAT (1 w after first dose)
32.	Kishimoto M. [78]	2022	Case series	2	COVID-19 mRNA vaccine	One SAT case with masive liver dysfunction
33.	Huo J. [79]	2022	Case report	1	COVID-19 vaccination	SAT (after 1 w)

Abbreviations: SAT = subacute thyroiditis; SL = silent thyroiditis; w = week; d = days; h = hour; NA = not available. * After the first dose (the patient had a prior diagnosis of Hashimoto’s thyroiditis). ** After the second dose; the patients were sisters. *** The diagnosis was established as a post-operative pathological diagnosis. **** Also confirmation based on fine-needle aspiration cytology. ***** Two of the three females were in the breastfeeding period. ****** The study is also mentioned in Table 1 since it includes both types of subacute thyroiditis (vaccine-induced and virus-induced).

## Data Availability

Not applicable.

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
