# Peer review of "The New Entity of Subacute Thyroiditis amid the COVID-19 Pandemic: From Infection to Vaccine"

_diagnostics, 2022, doi:10.3390/diagnostics12040960_

Round 1
Reviewer 1 Report
Whilst this is a good summary of the literature, I am not sure what this adds further, as there are a number of summaries on this already. This is not a systematic review ( it has not been prospectively registered), and the studies have not been assessed for bias. So it is more of a narrative review of the literature, which in itself can lead to bias. As such, I would be happy for the academic editor to make the final decision- however, the English language and style needs significant editing for grammar and vocabulary - the narrative structure itself flows well but the poor english makes the article hard to read. Furthermore, I would like the authors to clearly state that the benefits of vaccines far outweigh any transient issues with the thyroid
Reviewer 2 Report
A line 59 and some more lines below use the term "thyrotoxicosis" in the sense of "hyperthyroidism" which is wrong. Thyrotoxicosis is just an extreme degree of hyperthyroidism, but not a synonym. In de Quervain as a rule hyperthyroidism is by far less pronounced compared to Plummer's disease of von Basedow-Graves disease, that's why I recommend to swap for "Hyperthyroidism" . In endocrine section few syndromes and disorders are listed by commas, one by one, although some of them are not endocrine or not always endocrine ("sleep disorders" etc). This mess creates the impression of the lack of systematic approach. Overall impression decreases because authors applied many efforts to register and mention various cases described, but used much less efforts in order to analyze the material collected. Almost absent is a review of pathogenesis of the thyroid involvement in post COVID-19, no mention of some established mechanisms, like molecular mimicry, polyclonal immunostimulation, mechanisms of direct viral involvement of thyrocytes etc. An article could embed more pathophysiology, the personality of authors is not manifested in it enough or intentionally erased because of their modesty (which of course is a good feature of a person, but not a pillar of creativity...) .
Great disadvantage of a paper is that it was intentionally limited with English literature. The minority of mankind are English speakers, so the approach used by authors looks like blinkers on the eyes of the horses. Alas, very interesting papers and appropriate papers in French, Spanish, Italian, Russian, Chinese, Japanese, Polish and other languages directly related to the topic were out of scope of author's attention. Although, reviewer's experience shows that many Romanians speak these languages (especially Romanсe and Slavic ones) and Google translator speaks fare all languages to those readers who are not blinkered. Hence, the final product looks like "A Clobe of British Empire" which is funny.
Round 2
Reviewer 1 Report
I am happy with the changes
Reviewer 2 Report
You did your best.